# TET2-mediated epigenetic reprogramming of breast cancer cells impairs lysosome biogenesis

Audrey Laurent, Thierry Madigou, Maud Bizot, Marion Turpin, Gaëlle Palierne, Elise Mahé, Sarah Guimard, Raphaël Métivier, Stéphane Avner, Christine Le Péron, Gilles Salbert

**Methylation and demethylation of cytosines in DNA are believed to act as keystones of cell-specific gene expression by controlling the chromatin structure and accessibility to transcription factors. Cancer cells have their own transcriptional programs, and we sought to alter such a cancer-specific program by enforcing expression of the catalytic domain (CD) of the methylcytosine dioxygenase TET2 in breast cancer cells. The TET2 CD decreased the tumorigenic potential of cancer cells through both activation and repression of a repertoire of genes that, interestingly, differed in part from the one observed upon treatment with the hypomethylating agent decitabine. In addition to promoting the establishment of an antiviral state, TET2 activated 5mC turnover at thousands of MYC-binding motifs and down-regulated a panel of known MYC-repressed genes involved in lysosome biogenesis and function. Thus, an extensive cross-talk between TET2 and the oncogenic transcription factor MYC establishes a lysosomal storage disease–like state that contributes to an exacerbated sensitivity to autophagy inducers.**

## Introduction

Cell-specific gene expression programs are sustained by epigenetic landscapes that are established by enzymes targeting histones and DNA. Accordingly, genome-wide epigenomic rewiring is associated with acquisition of new cellular identities during development [1]. Cancer cells, although maintaining a cell-of-origin epigenomic imprint, acquire specific epigenomic features, some of which are common among different cancer types [2, 3, 4]. One such cancer-associated epigenetic feature is the so-called CpG island methylator phenotype (CIMP) in which hypermethylation of a substantial number of CpG islands (CGIs) that surround transcription start sites (TSSs) is found to be associated with low gene expression of the corresponding genes [3, 5]. In agreement with the idea that CGI methylation can occur at similar positions in various cancers, CGIs

from the clustered proto-cadherin (PCDH) tumor suppressor locus are frequently found methylated in breast, Wilms', cervical, colorectal, gastric, and biliary tract cancers [6, 7, 8, 9]. Although the existence of a breast cancer CIMP (B-CIMP) has been debated, a phenotype comparable with colon cancer and the glioma CIMP has been evidenced and suggested to be prevalent in the estrogen receptor $\alpha$ (ER$\alpha$)– and progesterone receptor (PR)–positive luminal subtypes of breast tumors [10]. Such an association of B-CIMP with the ER$\alpha$ and PR status was later confirmed, but no correlation with tumor size, lymph node invasion, and metastasis could be evidenced [11]. Although it is not precisely known what triggers the CIMP, a decrease in the activity of ten eleven translocation (TET) enzymes has been documented in various cancers [12] and linked to the occurrence of the CIMP in leukemia [13] and colorectal cancer [14]. TETs are 2-oxoglutarate/Fe$^{2+}$-dependent dioxygenases that iteratively oxidize 5-methylcytosine (5mC) into 5-hydroxymethylcytosine (5hmC), 5-formylcytosine (5fC), and 5-carboxylcytosine (5caC), with 5fC and 5caC being replaced by unmodified cytosines through the successive action of the DNA glycosylase TDG and the base excision repair machinery [15]. Consistent with a role in maintaining a hypomethylated state in CpG-rich regions, the TET1 knockout in mouse embryonic stem cells (mESCs) leads to CGI hypermethylation, suggesting that the CIMP could indeed be caused by reduced TET activity in cancer cells [16]. However, TET1 engagement at promoters can also repress gene expression by favoring the recruitment of PRC2, a complex mediating H3K27 methylation [16]. This dual action of TET enzymes towards gene regulation suggests that enforcing TET activity in cancer cells could provide additional benefits compared with epigenetic drugs commonly used to inhibit DNA methylation such as 5-azacytidine and 5-aza-deoxycytidine (decitabine). Here, we genetically engineered MCF-7 cells, a model of low-metastatic luminal breast cancer cells that express both ER$\alpha$ and PR transcription factors and whose growth is partly dependent on estrogen supply [17], to artificially modify their epigenome. Enforcing TET2 activity reduced the tumorigenic potential of MCF-7 breast cancer cells and triggered an antiviral state and a lysosomal storage disease–like phenotype that predisposed cells to death.

Université Rennes 1, CNRS UMR6290, Institut de Génétique et Développement de Rennes, Campus de Beaulieu, Rennes, France

Correspondence: christine.le-peron@univ-rennes1.fr; gilles.salbert@univ-rennes1.fr

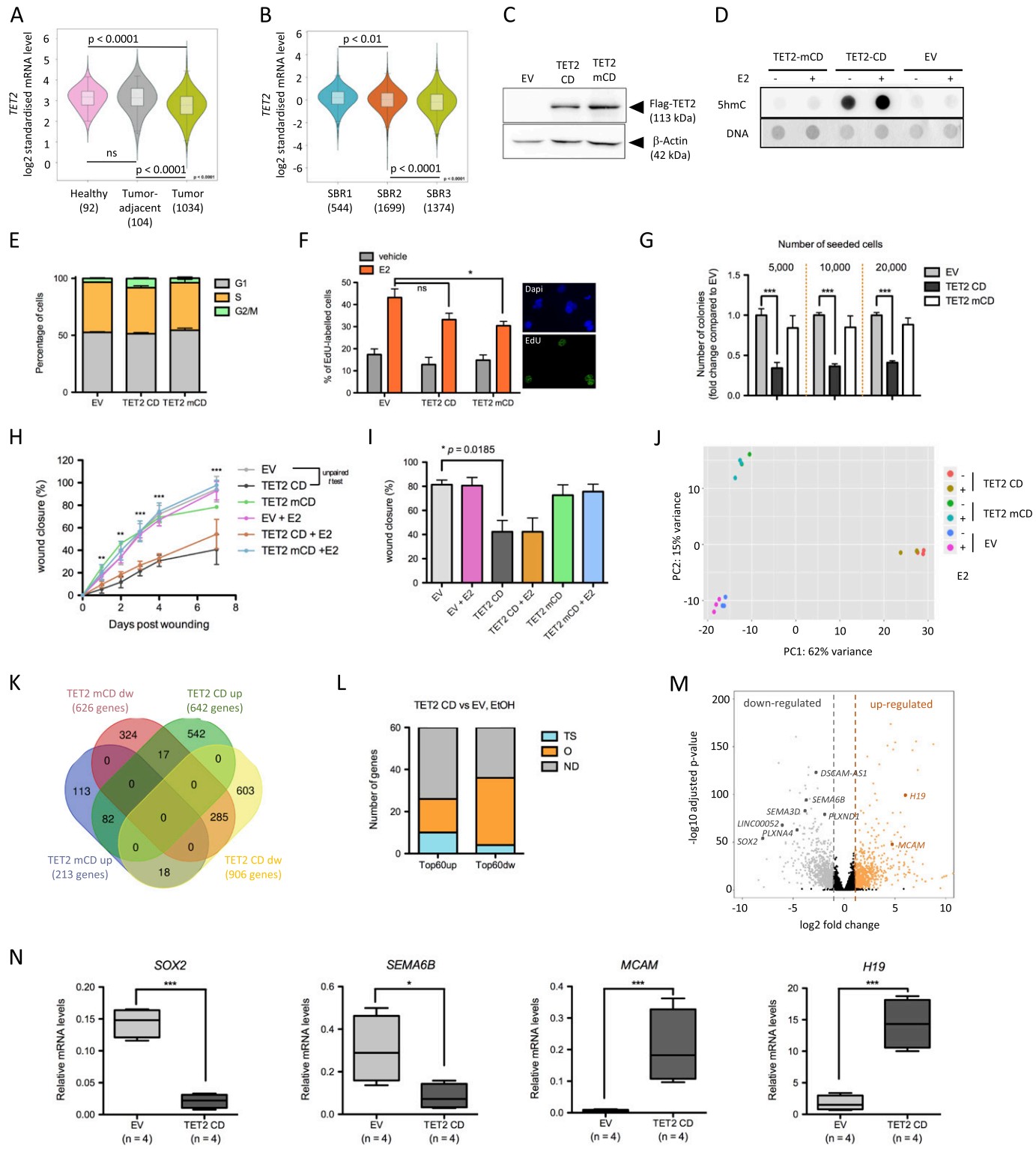

**Figure 1. TET2 CD expression reduces the tumorigenic potential of MCF-7 cells.**
**(A)** TET2 expression in healthy tissue, tumor adjacent, and tumors of BRCA patients. **(B)** TET2 expression in BRCA tumors according to their Scarff Bloom and Richardson (SBR) grade status. In (A) and (B), plots were generated with Breast Cancer Gene-Expression Miner v4.5. **(C)** Western blot detection of Flag-TET2 CD and TET2 mCD in MCF-7 clones. **(D)** Dot blot analysis of 5hmC levels in 500 ng of genomic DNA from empty vector (EV), TET2 CD, and TET2 mCD cells treated or not with estradiol (E2). DNA was stained with methylene blue. **(E)** Flow cytometry of propidium iodide–labeled EV, TET2 CD, and TET2 mCD cells. Bar graphs show the distribution of cells in G1, S, and G2/M phases of the cell cycle (n = 5). **(F)** Bar graph representation of EdU labeling of S-phase cells in the presence or absence of E2 (n = 3). Images show DAPI staining and EdU labeling of a representative microscopic field of EV cells treated with E2. **(G)** Anchorage-independent clonogenic assay of MCF-7–derived clones grown in soft agar for 4 wk. Bar graphs

# Results

## Active TET2 CD alters the tumorigenic potential of MCF-7 cells

Examination of TET gene expression in breast cancer patients revealed that although TET3 mRNA levels were higher in tumors and positively correlated with tumor progression, both TET1 and TET2 expression were diminished in tumor cells, and TET2 expression decreased with tumor progression more significantly than TET1 (Figs 1A and B and S1A). We thus chose to ectopically express TET2 in breast cancer cells and transfected MCF-7 cells with an expression vector for the Flag-tagged active murine TET2 CD and as controls, with a catalytically dead mutant (H1304Y and D1306A) TET2 mCD (18) or an empty vector (EV). Clones were isolated in the presence of Geneticin and analyzed for expression of Flag-TET2 (Fig 1C) as well as for their enrichment in 5hmC (Fig 1D). Consistent with a role of TET2 in 5mC oxidation, increased 5hmC levels were observed in TET2 CD cells but not in TET2 mCD cells. Although TETs have been suggested to regulate the cell cycle (19), no significant changes in the distribution of the cells in the different cell cycle phases were noticed by flow cytometry analysis (Fig 1E). In addition, S-phase entry was still enhanced by estradiol in TET2 CD and mCD cells, as assessed by EdU labeling (Fig 1F). Conversely, TET2 CD cells were less prone to grow as colonies in an anchorage-independent growth assay (Fig 1G) and to migrate in a wound healing assay (Fig 1H and I). Collectively, these data show that enforcing active TET2 CD expression mitigates the tumorigenicity of MCF-7 cells and are consistent with previous work showing that both TET1 and TET2 reduce tumor growth in xenograft mice (20, 21).

To explore further the impact of TET2 CD and mCD expression in the MCF-7 clones, their transcriptome was established through Illumina sequencing of poly-dT-captured mRNAs. Principal component analysis of the 500 most expressed genes indicated a major reconfiguration of RNA pol II–mediated transcription in TET2 CD cells and, to a lower extent, in TET2 mCD cells, with little impact of estradiol (Fig 1J). Comparison of TET2 CD and mCD differentially expressed genes (DEGs, fold change [FC] ≥ 2 and adjusted *P*-value ≤ 0.05 when compared with an EV) evidenced that the catalytic activity of TET2 was not required for all the transcriptional changes observed (Fig 1K). Indeed, although the number of regulated genes was lower in the case of TET2 mCD, 12.7% (82 genes) of the 642 genes activated by TET2 CD were also activated by the expression of TET2 mCD and 31.4% (285 genes) of the 906 genes repressed by TET2 CD were also repressed by TET2 mCD. This suggests that TET2 CD can exert 5mC oxidation–independent functions that are more prominent for gene repression than for gene activation and also indicates that 5mC oxidation per se is an important determinant of TET2 CD–mediated gene regulation. Close examination of the top 60

differentially regulated genes in TET2 CD cells versus EV cells revealed that a rather similar proportion of described oncogenes (26.7%), including the long noncoding RNA (lncRNA) *H19* ((22); Fig 1L and M) and tumor suppressor genes (16.7%), were up-regulated (Fig 1L and Table S1). Conversely, 56.7% of the top 60 down-regulated genes were known oncogenes (Fig 1L and Table S1), among which are the lncRNAs *DSCAM-AS1*, a luminal marker involved in breast cancer progression (23, 24), and *LINC00052*, an anchorage-independent growth promoter ((25); Fig 1M). Notably, four genes encoding semaphorins (SEMA) of the oncogenic type and their receptors (PLXN) were among the top 60 down-regulated genes in TET2 CD cells (Fig 1M). *SEMA3D* and *SEMA6B*, as well as the receptors *PLXNA*4 and *PLXND1*, act as oncogenes favoring cell growth and migration (26, 27, 28). Interestingly, the mixed oncogene/tumor suppressor *MCAM*, which is induced through interaction of the tumor suppressor–type semaphorin SEMA3A with its receptor NPR1 and silenced by promoter methylation in MCF-7 cells (29, 30), was among the top 60 TET2 CD up-regulated genes (Fig 1M). In addition, a dramatic reduction in expression of the major oncogene *SOX2* (31, 32) was evidenced (Fig 1M). Additional TET2 CD clones were tested by RT-qPCR that consistently showed *SOX2* and *SEMA6B* down-regulation, and up-regulation of *MCAM* and *H19* (Fig 1N). Except for *SOX2*, gene expression changes for *SEMA6B*, *MCAM*, and *H19* in MCF-7 cells treated with the hypomethylating agent decitabine ((33), GSE74036) were similar to those observed upon TET2 CD expression, validating the hypothesis that these genes are regulated by DNA methylation (Fig S1B). Finally, consistent with a lower tumorigenicity of TET2 CD cells, interrogation of proteomic data obtained from a panel of various breast cancer cell lines (34) showed that TET2 CD cells activated genes encoding proteins enriched in the proteome of low tumorigenic cells and repressed genes encoding proteins that accumulate in highly tumorigenic cells (Fig S1C–F). Similar gene expression changes were also observed in decitabine-treated MCF-7 cells (Fig S1G). As a whole, this set of data indicates a lower aggressiveness of TET2 CD cells, which correlates with a down-regulation of master regulators of cell growth and migration, likely contributing to their lower tumorigenic potential.

## TET2 CD triggers both activation and repression of CGI promoters

To test whether TET2 CD expression could reprogram CGIs, we next mapped 5hmCpGs genome-wide using a base resolution method relying on selective chemical labeling (SCL) coupled to exonuclease digestion (SCL-exo, (35)). Average profiles of SCL-exo signal centered on 28,691 hg19 CGIs showed enrichment in 5hmCpGs specifically at CGIs from TET2 CD cells (Fig 2A). Consistent with 5hmC occurring at methylated CpGs, average 5hmC enrichment was more

---

indicate the number of colonies (mean ± SEM, n = 6) for three initial seeding densities. **(H, I)** Wound healing assay showing delayed migration of TET2 CD cells. **(H, I)** Wound closure was quantified at different time points after wounding, and data are shown as mean ± SEM (n = 6) for all time points from a single experiment (H) or as mean ± SEM (n = 3) for day 4 samples from three independent experiments (I). **(J)** Principal component analysis of RNA-seq samples based on the 500 most expressed genes in each sample. **(K)** Venn diagram showing the overlap between the lists of up- or down (dw)-regulated genes in TET2 CD and TET2 mCD cells compared with EV cells in the absence of E2 (FC ≥ 2). **(L)** Literature-based annotation of the top 60 up- and down-regulated genes without E2 in TET2 CD cells compared with EV cells (TS: tumor suppressor, O: oncogene, ND: not determined). **(M)** Volcano plot visualization (*P*-values versus Log$_2$FC) of differentially expressed genes (DEGs) between TET2 CD and EV cells. Genes described in text have been highlighted. Dashed vertical lines indicate −1 and +1 log$_2$ fold changes. **(N)** Box plot representation of RT-qPCR measurements of *SOX2*, *SEMA6B*, *MCAM*, and *H19* RNAs in four independent EV and TET2 CD clones (mean ± SEM, n = 4).

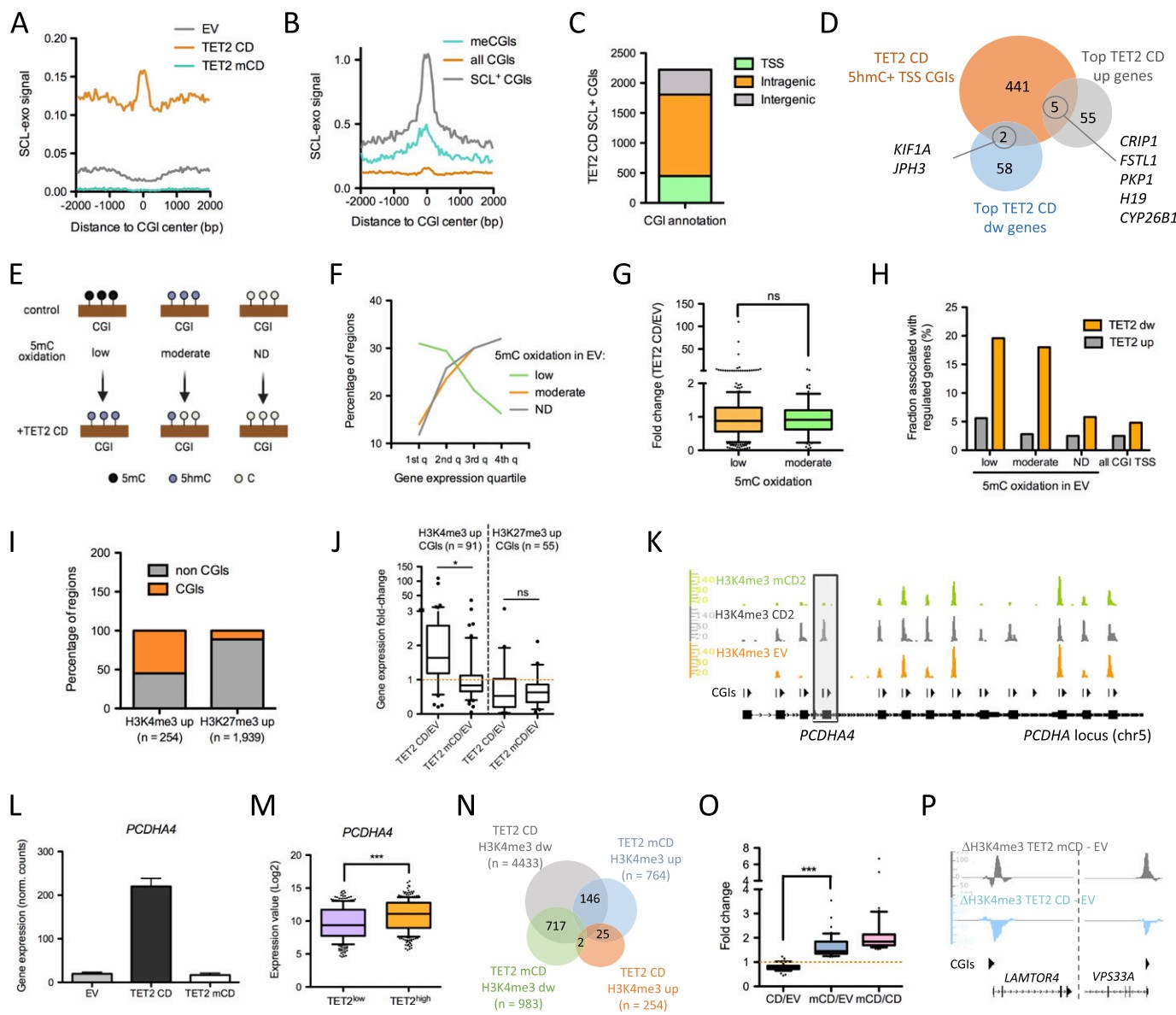

**Figure 2. CGI reprogramming by TET2 CD.**
**(A)** Average profile of 5hmC levels (selective chemical labeling-exo signal) in empty vector (EV), TET2 CD, and TET2 mCD cells, centered at hg19 CGIs (n = 28,691). **(B)** Average profile of 5hmC levels in TET2 CD cells centered at all hg19 CGIs, at CGIs methylated in MCF-7 cells (n = 2,330), or at selective chemical labeling–positive CGIs in TET2 CD cells (n = 2,224). **(C)** Called 5hmC-positive CGIs in TET2 CD cells were annotated as intergenic, intragenic, or within ±500 bp of a transcription start site (TSS). **(D)** Venn diagram visualization of the overlap between top 60 DEGs (up and down) and 5hmC-positive TSS-associated CGIs in TET2 CD cells. **(E)** Classification of TSS-CGIs as a function of 5hmC variation between EV and TET2 CD cells. Low–5mC oxidation TSS-CGIs have no detectable 5hmC signals in EV cells and a gain in TET2 CD cells. Moderate–5mC oxidation TSS-CGIs have detectable 5hmC signals in EV and a decreased signal in TET2 CD cells. TSS-CGIs with no detectable (ND) 5hmC signals either in EV cells or in TET2 CD cells, which are supposed to be protected from DNA methylation. **(F)** Association of the classified TSS-CGIs with gene expression quartiles (the first quartile including the lowest expression levels to the fourth quartile with the highest expression levels). **(G)** Expression fold change (TET2 CD/EV) of genes associated with low– or moderate–5mC oxidation TSS-CGIs. **(H)** Fraction of low–, moderate–, and ND–5mC oxidation TSS-CGIs associated with up- and down-regulated genes in TET2 CD cells compared with EV cells. **(I)** CGI association of H3K4me3-up or H3K27me3-up regions in TET2 CD cells versus EV cells. **(J)** Expression fold change of genes associated with H3K4me3-up or H3K27me3-up CGIs in TET2 CD or TET2 mCD versus EV cells. **(K)** Integrated Genome Browser (IGB, https://bioviz.org/) visualization of H3K4me3 ChIP-seq signals at the *PCDHA* locus in EV and TET2 CD and TET2 mCD cells. CGI positions are indicated. **(L)** *PCDHA4* gene expression levels in EV, TET2 CD, and mCD cells (normalized RNA-seq read counts, mean ± SEM, n = 3). **(M)** *PCDHA4* mRNA levels in breast cancer tumors (TCGA BRCA panel) classified into TET2$^{low}$ (first quartile n = 301) and TET2$^{high}$ (fourth quartile, n = 299; samples with a value of 0 were discarded). **(N)** Venn diagram indicating the overlap between regions that gained (up) or lost (dw) H3K4me3 in TET2 CD and mCD cells compared with EV cells. **(O)** Expression fold change of the 42 genes showing opposite H3K4me3 signal variations in TET2 CD and mCD cells compared with EV cells, and they were up-regulated more than 1.2-fold in TET2 mCD versus EV cells and more than 1.5-fold in TET2 mCD versus TET2 CD. **(P)** IGB visualization of the differential H3K4me3 signal at the *LAMTOR4* and *VPS33A* loci.

pronounced at CGIs identified as methylated in MCF-7 cells ((36); Fig 2B). We then isolated 2,224 CGIs having at least four hydroxymethylated CpGs in TET2 CD cells. These 5hmC-positive CGIs showed a high SCL-exo signal (Fig 2B), and 20% (448) of them were located within 500 bp of a TSS (Fig 2C). However, these TSS-associated hydroxymethylated CGIs were found to associate poorly with highly regulated genes. Indeed, among the top 60 TET2 CD–regulated genes, only 5 up- and 2 down-regulated genes had a TSS associated with a 5hmC+ CGI (Fig 2D). CGIs including TSSs, thereafter designated TSS-CGIs, were next classified into three groups (Figs 2E and S2A): (*i*) those having no detectable 5hmC in EV cells and gaining 5hmC in TET2 CD cells (low 5mC oxidation in EV cells, n = 1,034); (*ii*) those having readily detectable 5hmC in EV cells and losing 5hmC in TET2 CD cells (n = 95), likely because of further oxidation of 5hmC by the expressed TET2 CD (moderate oxidation in EV cells); and (*iii*) those showing no signal at all in the EV and TET2 CD cells (n = 10,839) and which may correspond to CGIs subjected to a very high 5mC oxidation rate or being fully protected from methylation (no modification). Among low-oxidation TSS-CGIs, only 34.5% were associated with a gene with detectable reads in RNA-seq, compared with 75% and 74% for the other two sets of TSS-CGIs, indicating that TET2-mediated gain in 5hmC at TSS-CGIs essentially targeted silent genes that did not get activated when demethylated. Accordingly, low-oxidation TSS-CGIs tended to associate with the lowest expression quartiles of EV cell genes, whereas moderate-oxidation TSS-CGIs and unmodified TSS-CGIs were more associated with the highest expression quartiles (Fig 2F). Fold changes (TET2 CD versus EV) of genes associated with low- and moderate-oxidation TSS-CGIs were not significantly different (Fig 2G), although their variance was different ($P < 0.0001$). This was correlated to a higher representation of activated genes in low-oxidation TSS-CGIs, whereas low- and moderate-oxidation TSS-CGIs equally associated with down-regulated genes (Fig 2H). Altogether, these data indicate that 5mC oxidation dynamics at TSS-CGIs per se is not a robust predictor of the directionality of gene expression changes.

To further explore CGI chromatin remodeling upon TET2 CD expression, changes in the distribution of the active promoter mark H3K4me3 and the PRC2-mediated repression mark H3K27me3 were investigated by ChIP-seq in MCF-7 clones. Quite few genomic regions gained H3K4me3 (n = 254) in TET2 CD cells compared with regions gaining H3K27me3 (n = 1,939). However, 55.1% (140 of 254) of these H3K4me3-up regions overlapped with CGIs compared with 11.3% (220 of 1,939) of H3K27me3-up regions (Fig 2I). Contrary to 5mC oxidation, changes in the levels of histone modifications reflected gene expression changes in TET2 CD cells versus EV cells, with CGIs gaining H3K4me3 being associated with activated genes and CGIs gaining H3K27me3 with repressed genes (Fig 2J). In addition, genes gaining H3K4me3 in TET2 CD cells were not activated in TET2 mCD cells, whereas genes gaining H3K27me3 in TET2 CD cells tended to be repressed in TET2 mCD cells (Fig 2J), suggesting that gene repression also occurred in the absence of 5mC oxidation, probably through protein–protein interaction between PRC2 and the catalytic domain of TET2. However, as verified by ChIP-qPCR on 2 down-regulated genes (*LHX2* and *JPH3*), the levels of both gene repression and H3K27 methylation were much lower in the absence of TET2 catalytic activity (Fig S2B–D). This was in accordance with previous studies suggesting that an active TET2 catalytic domain was

required for deposition of H3K27me3 at CGIs (37). Interestingly, decitabine was not able to repress *LHX2* and *JPH3* (Fig S2E), indicating that TET2–PRC2 interaction in combination with DNA demethylation is probably required for repression of these genes. Consistent with these observations, mRNA levels of both *LHX2* and *JPH3* were significantly higher in breast cancer patient samples with low TET2 mRNA levels than in high-TET2 samples (Fig S2F).

As an example of TET2 CD–mediated gene activation, a subset of TSS-CGIs from the breast cancer-methylated *PDCHA* locus (6) selectively gained H3K4me3 in TET2 CD cells in correlation with the activation of the associated genes as shown for *PCDHA4* (Fig 2K and L), *PCDHA3*, *PCDHA9*, and *PCDHA10* (Figs 2K and S2G). Interrogation of TCGA RNA-seq data from breast cancer samples supported the data obtained from MCF-7 cells, showing that *PCDHA4* expression levels are positively correlated with *TET2* expression in patients (Fig 2M). To unveil a potential dominant-negative function of TET2 mCD, we next focused on regions showing opposite H3K4me3 variations. Consistent with correlated gene repression and uncorrelated gene activation between TET2 CD and mCD cells (Fig 2J), 73% (717 of 983) of H3K4me3-down regions in TET2 mCD cells also lost H3K4me3 in TET2 CD cells, whereas a limited overlap was observed for H3K4me3-up regions (3.2%, 25 of 764, Fig 2N). Highlighting a potential dominant-negative function of TET2 mCD at a subset of sites, 19% (146 of 764) of the H3K4me3-up regions in TET2 mCD compared with EV cells were called as H3K4me3-down in TET2 CD cells (Fig 2N). Among the 136 annotated genes associated with these opposite H3K4me3 changes, 42 had both a TET2 mCD/TET2 CD fold change above 1.5 and a TET2 mCD/EV fold change above 1.2 (Fig 2O). GO annotation for cellular components of these 42 genes with Pantherdb ((38); http://pantherdb.org/) indicated a unique annotation for intracellular organelles (GO:0043229) with a 1.5-fold enrichment (FDR = 4.50 × $10^{-02}$). In particular, from this list of 42 genes, six genes were involved in lysosome biogenesis and autophagy (*ATP6V0A2*, *LAMTOR4*, *PRKAB1*, *SLC15A4*, *SPPL3*, and *VPS33A*, Fig 2P).

## TET2 CD alters 5mC oxidation and H3K4me3 levels at MYC-binding sites

Because DNA methylation/demethylation can influence transcription factor binding to DNA (39, 40), high-resolution SCL-exo data were next interrogated for enriched transcription factor–binding sites (TFBSs) in TET2 CD cells. Iterative clustering of TET2 CD and EV 5hmCpGs using the heatmap clustering tool of Cistrome (41) isolated a set of CpGs (n = 24,008) with high 5hmC enrichment in TET2 CD cells compared with EV cells (Fig 3A), and another set of CpGs (n = 18,404) with the opposite enrichment pattern and most likely corresponding to 5hmCpGs undergoing superoxidation in TET2 CD cells (Fig S3A). Analysis of these two populations of CpGs with TFmotifView (42) revealed a high enrichment ($P = 0$ for CpGs gaining 5hmC and $P = 6.3 × 10^{-272}$ for CpGs losing 5hmC) in the MYC-binding E box CACGTG motif (Figs 3B and S3B). This is consistent with the observation that 72% of the MYC-bound TSSs are also engaged by TET2 in HEK293T cells (43). MYC ChIP-seq data (ENCODE, SRR575112.1) obtained from serum-fed MCF-7 cells were next used to generate a list of MYC-binding sites. These sites showed higher oxidation of 5mC (as reflected by an increase in differential 5hmC levels) in TET2 CD cells than in EV cells and a lower one in TET2 mCD

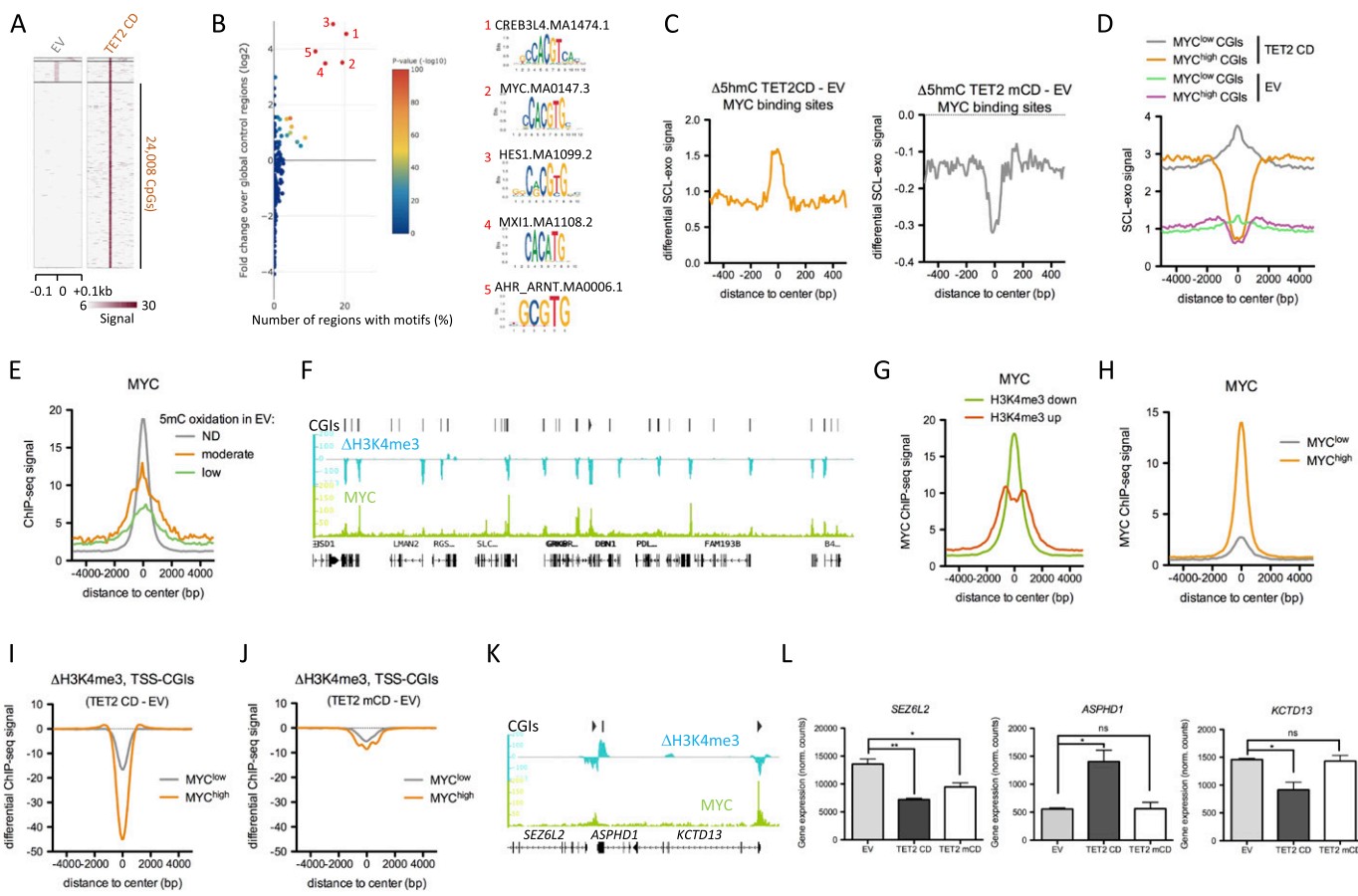

**Figure 3. TET2 CD targets MYC-binding sites.**
**(A)** Identification of CpGs gaining 5hmC in TET2 CD cells versus empty vector (EV) cells by heatmap clustering. **(B)** Enrichment of transcription factor binding motifs at CpGs gaining 5hmC in TET2 CD cells. Graph was generated with TFmotifView. **(C)** Differential selective chemical labeling-exo signal between TET2 CD and EV cells (left panel) or TET2 mCD and EV cells (right panel) at MYC-binding sites in MCF-7 cells (SRR575112.1). **(D)** Selective chemical labeling-exo signal at MYC$^{low}$ and MYC$^{high}$ CGIs in EV and TET2 CD cells. **(E)** Average MYC enrichment at CGIs classified according to their 5mC oxidation rate. **(F)** IGB representation of MYC ChIP-seq signal and differential H3K4me3 signal between TET2 CD and EV cells at a 400-kb region of chromosome 5. **(G)** Average MYC ChIP-seq signal at H3K4me3-up or -down regions in TET2 CD cells compared with EV cells. **(H)** Average MYC ChIP-seq signal at MYC$^{high}$ and MYC$^{low}$ transcription start site-CGIs. **(I, J)** Differential H3K4me3 signal (I: TET2 CD - EV, J: TET2 mCD - EV) at MYC$^{high}$ and MYC$^{low}$ transcription start site-CGIs. **(K)** IGB snapshot of MYC ChIP-seq signal and H3K4me3 differential signal (TET2 CD - EV) at the *SEZ6L2*, *ASPHD1*, and *KCTD13* loci. **(L)** *SEZ6L2*, *ASPHD1*, and *KCTD13* mRNA levels in TET2 CD, mCD, and EV cells (normalized RNA-seq read counts, mean ± SEM, n = 3).

cells, consistent with a dominant-negative effect of the catalytic dead mutant (Fig 3C). CGIs were next categorized into MYC$^{high}$ and MYC$^{low}$ subsets based on MYC ChIP-seq data from MCF-7 cells (Fig S3C). MYC$^{high}$ CGIs were depleted in 5hmC, whereas MYC$^{low}$ sites were enriched in 5hmC (Figs 3D and S3C). A large fraction of MYC-binding sites were associated with TSS-CGIs (Fig S3D), and consistent with an inhibitory role of DNA methylation in MYC binding to DNA (39, 44), a higher engagement of MYC was found at TSS-CGIs with moderate 5mC oxidation than at TSS-CGIs with low 5mC oxidation (Fig 3E). The MYC transcription factor plays pleiotropic roles in cancer cells, both activating and repressing transcription, and the genome-wide binding of MYC to E boxes is biased towards H3K4me3-enriched sites (45, 46). Of note, as exemplified in Fig 3F, a large fraction of MYC-bound TSS-CGIs showed decreased levels of H3K4me3 in TET2 CD cells compared with EV cells, although other genomic regions showed mixed behaviors with CGIs either gaining H3K4me3 or showing no change (Fig S3E). Accordingly, MYC binding was detected at the center of H3K4me3-down regions and slightly off the center of H3K4me3-up regions, suggesting a strong relationship between promoter activity and MYC binding (Fig 3G). TSSs from TSS-CGIs were next split into MYC$^{low}$ and MYC$^{high}$ subsets and analyzed for variation in H3K4me3 in TET2 CD and TET2 mCD cells compared with EV cells (Fig 3H). Results showed that the degree of H3K4me3 loss at TSSs from TSS-CGIs in TET2 CD cells was correlated to the level of MYC binding in MCF-7 cells (Fig 3I and J). Such a decrease in H3K4me3 levels was not observed in TET2 mCD cells, indicating a requirement for an active catalytic domain. As an example, the *SEZ6L2* and *KCTD13* TSS-CGIs, which both bind MYC in MCF-7 cells, lost H3K4me3 in TET2 CD cells compared with EV cells, whereas the *ASPHD1* TSS-CGI was not bound by MYC and gained H3K4me3 (Fig 3K). Consistent with these observations, *SEZ6L2* and *KCTD13* showed a lower expression and *ASPHD1* a higher expression in TET2 CD cells (Fig 3L). Collectively, these data are in strong support of an increased engagement of MYC at TSS-CGI–binding sites upon 5mC oxidation by TET2 CD, leading to lower transcription of the associated genes.

## Decitabine and TET2 CD induce distinct cell reprogramming

Cancer alterations in DNA methylation can be counteracted by using DNA methyltransferase inhibitors such as decitabine and 5-azacytidine (47). These drugs have been shown to promote tumor regression in hematological malignancies (48, 49, 50) and are under intensive investigation in solid tumors (51, 52). Combining HDAC inhibitors and DNA hypomethylating agents further reduced proliferation of lung cancer cells by decreasing MYC levels and reversing immune evasion (52). Interestingly, DNMT inhibitors activate antiviral response genes through production of dsRNA from endogenous retroviral elements (ERVs), promoting apoptosis and immune checkpoint therapy in epithelial cancer cells (53, 54, 55). To compare the respective impact of TET2 CD expression and decitabine in MCF-7 cells, we first extracted DEGs (FC ≥ 2 and FC ≤ 0.5, adjusted $P$-value ≤ 0.05) from public RNA-seq data of MCF-7 cells treated daily with 100 nM decitabine for 96 h (33). Consistent with the data obtained from other cell lines (33), the main outcome of decitabine in terms of gene regulation in MCF-7 cells was activation. Indeed, only 4.7% (80 of 1704) of the DEGs were down-regulated by decitabine treatment (Fig 4A). This was in striking contrast to TET2 CD DEGs, which showed 58.5% (906 of 1,548) of down-regulated genes (Fig 4A). In addition, the set of DEGs poorly overlapped between decitabine treatment and TET2 CD, with only 129 up-regulated genes in common, including *MCAM* (Fig 4B and C). Conversely, *DSCR8*, an lncRNA known to activate WNT/$\beta$-catenin signaling in hepatocellular carcinoma (56), was confirmed to be massively induced by decitabine in MCF-7 cells by RT-qPCR (Fig 4B and C). These data suggest that decitabine-induced and TET2 CD–induced transcriptional changes differ substantially.

Next, GO annotation of decitabine and TET2 CD–induced genes revealed a TET2 CD–specific enrichment in antiviral response genes (Fig 4D and E). Such genes included pattern recognition receptors (PRRs) involved in viral RNA sensing (*MDA5*, *LGP2*, *RIG-1*, and *PKR*), transcription factors (*IRF9* and *STAT1*), and interferon-stimulated genes (ISGs), whereas none of the interferon genes were induced (Fig 4E and F), consistent with an already observed interferon-independent activation of ISGs (57). Of note, the four *OAS* genes which are involved in RNAse L activation through synthesis of 2′-5′-oligoadenylate (58) were highly induced in TET2 CD cells, whereas the RNAse L inhibitor ABCE1 showed a moderate twofold increase (Fig 4F). As already described in various cancer cell types, decitabine activated a subset of these genes (Fig 4F). TET2 CD induction of *DDX58*, *OAS2*, and *IFIT1* was confirmed by RT-qPCR, and these genes were further activated upon decitabine treatment (Fig 4G). However, when looking at the correlation between TET2 expression and antiviral response genes in breast cancer patients, a positive correlation was found only for *EIF2AK2*, *MAVS*, *OTUD4*, *ABCE1*, and *RNase L* expression levels (Fig 4H). By contrast, expression of ISGs in patients strongly correlated with expression of PRRs but not with expression of *MAVS*, *OTUD4*, *ABCE1*, and *RNase L*. One possible explanation could be that tumor cells with high PPRs, MAVS, OTUD4, ISGs, and RNAse L are undergoing cell death and are thus counterselected. The antiviral state triggered by decitabine has been shown to associate with an increased transcription of endogenous retroviruses (ERVs, (54, 55)). Consistent with these studies, transcription of HERV-Fc1 was increased by decitabine in our cell lines,

but basal expression was higher in TET2 CD cells and in TET2 mCD cells (Fig 4I). Conversely, the LTR12C RNA levels were high in all conditions (Fig 4I). In agreement with these expression data, dot blot analysis of dsRNA levels did not show dramatic differences between cell clones (Fig 4J). Considering that antiviral genes were not induced in TET2 mCD cells, whereas HERV-Fc1 expression was increased, activation of the antiviral state by TET2 CD may require additional mechanisms. Knowing that (i) viral and ERV RNAs are methylated in cells (59, 60), (ii) RNA methylation decreases the antiviral response (61), and (iii) in ES cells, TET2 oxidizes 5mC in ERV RNAs (60); it is then possible that the high antiviral response in TET2 CD cells reflects a dual action of TET2 on both genomic DNA and RNA transcribed from repeated sequences. Such a scenario would be compatible with the additive effect observed when combining active TET2 CD and decitabine (Fig 4G and I). Of note, viral mimicry did not induce the death of TET2 CD cells, suggesting that the level of activation of the innate immune pathway remained below the threshold required for cell death commitment. Because RNAse L activation by 2′-5′-oligoadenylate is counteracted by the RNase L inhibitor RLI/ABCE1 (58, 62), we next tested the hypothesis that activation of the antiviral response pathway could sensitize TET2 CD cells to RNAse L–mediated cell death by transfecting siRNAs targeting ABCE1. Efficient knockdown of ABCE1 mRNA levels was observed, together with a massive induction of viral response genes in siRNA-transfected MCF-7 cells (Fig 4K). MTT assays of cells challenged with increasing concentration of ABCE1 siRNAs next revealed an increased ability of ABCE1 knockdown to induce cell death in TET2 CD cells compared with EV and TET2 mCD cells (Fig 4L). Collectively, these data indicate that enforced TET2 activity in MCF-7 breast cancer cells triggers a pre-activated antiviral state that predisposes cells to death induced by ABCE1 inactivation.

## TET2 regulates lysosome function

Innate immune response triggered by viral infection is associated with an RNase L–dependent autophagy of viral particles (63). Although autophagy was not a term enriched by gene ontology analysis of our RNA-seq data, a significant association of TET2 CD down-regulated genes (FC ≥ 2) with lysosome annotation was evidenced, and this association was even more pronounced when using a less stringent threshold of a 1.5-fold decrease (Figs 5A and S4A). As already suggested in Fig 2N–P, these data indicate that TET2 CD cells are endowed with altered lysosomal function. Down-regulation of *CLN3*, *CTSD*, and *NAGLU* in TET2 CD cells was further confirmed by RT-qPCR analysis, and expression of these three genes was shown to be anticorrelated with TET2 expression levels in the breast cancer TCGA cohort of patients (n = 1,218), validating our in vitro observations (Figs 5B and C and S4B and C). By using siRNAs targeting all three TETs, expression of *CTSD* and *CLN3* was shown to be specifically repressed by TET2 in MCF-7 and T47D breast cancer cells and, to a lower extent, in HEK293T kidney cancer cells (Fig S4D). Using the Pan-Cancer TCGA dataset gathering RNA-seq data from 11,060 patients, an anticorrelation between *TET2* expression and mRNA levels of lysosome proteins was confirmed for *CLN3*, *CTSD*, *CTSF*, *CTSZ*, *IFI30*, and *NAGLU*, suggesting TET2 might down-regulate lysosomal genes in various types of cancer (Fig S4E). To interrogate a possible impact of TET2 CD expression on lysosome activity, acidic

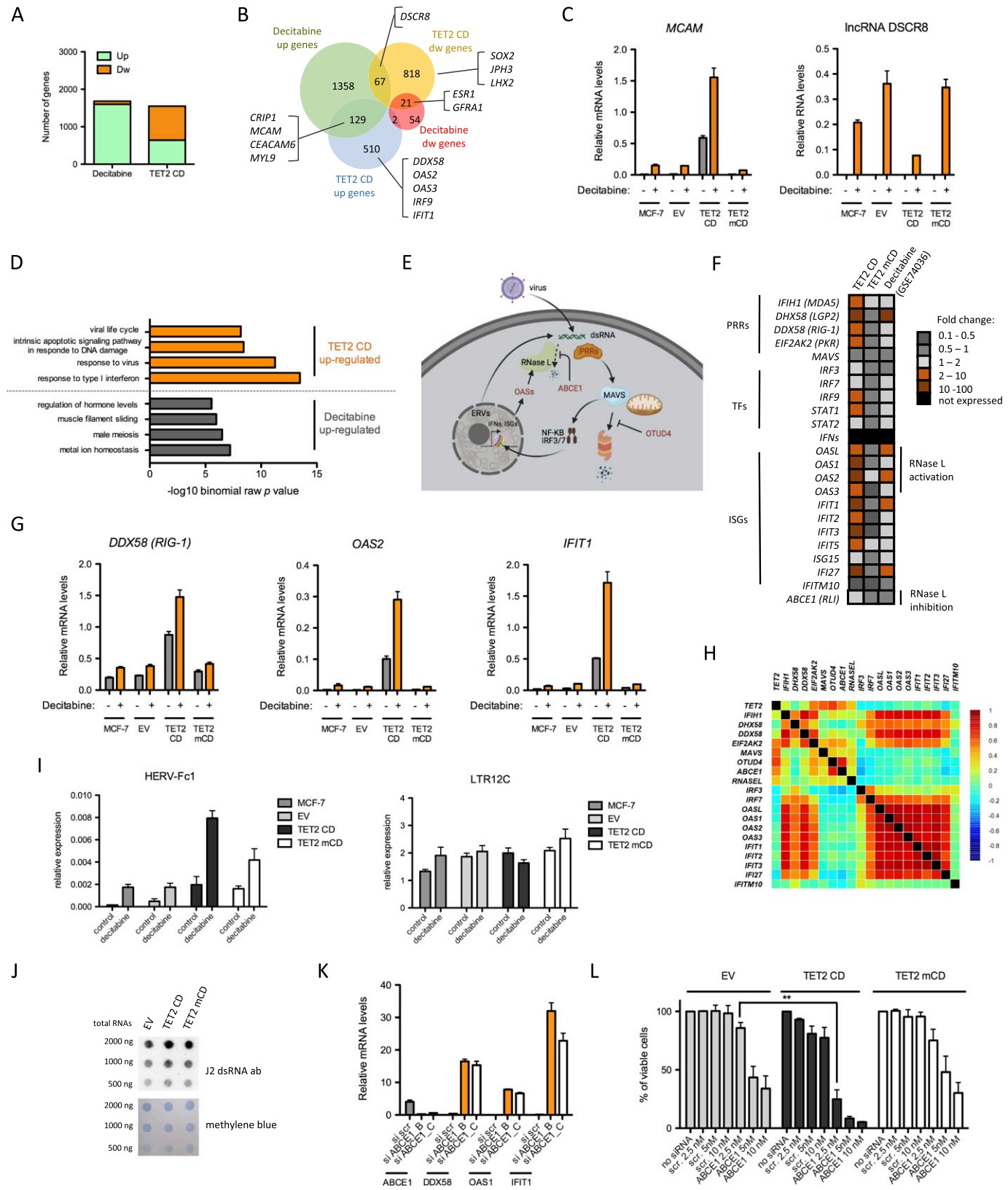

**Figure 4. Differential transcriptional rewiring between TET2 CD expression and decitabine treatment.**
**(A)** Distribution of up- and down-regulated genes in DEGs from decitabine-treated MCF-7 cells (GSE74036) and from TET2 CD cells versus empty vector (EV) cells.
**(B)** Overlap between DEG lists from decitabine-treated MCF-7 cells and from TET2 CD cells versus EV cells. A number of genes mentioned in text are highlighted. Note that

vesicles were next labeled with LysoTracker Red (64). Data showed that acidic vesicle size was higher in TET2 CD cells, indicating a potential engorgement of lysosomes (Fig 5D and E). Lysosomes often position next to the centrosome where they have a high probability to fuse with autophagosomes guided by molecular motors (65). Such a pericentrosomal positioning was obvious in the EV, TET2 CD, and TET2 mCD cells, but a fraction of TET2 CD cells showed lysosomes that were scattered around the nucleus (Fig 5D). Reduced levels of hydrolytic enzymes and mislocalized lysosomes have been observed in *CLN3*-mutant cells (66), and engorged lysosomes were described in *SNX14* mutants, causing cerebellar atrophy in humans (67). Consistent with these observations, *SNX14* mRNA levels were also reduced in TET2 CD RNA-seq data (Fig 5A).

Such an effect of TET2 CD expression on the activity of a large number of lysosomal genes was striking and suggestive of an alteration of a coordinated mechanism controlling lysosomal gene expression. It has been established that transcription factors from the basic helix–loop–helix leucine zipper (bHLH-ZIP) family such as TFEB and TFE3 coordinately activate lysosomal genes through binding of their basic domain to the coordinated lysosomal expression and regulation (CLEAR) motif (GTCACGTGAC) commonly found in the promoter of these genes (68, 69). In addition, an epigenetic mechanism involving MYC binding to the CLEAR motif (which contains the high-affinity MYC-binding site CACGTG) and recruitment of HDAC9 has been shown to antagonize the coordinated action of TFEB and TFE3 on lysosomal gene expression (70). Although MCF-7 cells did not express TFEB, they showed high levels of TFE3 and MYC mRNAs (Fig 5F), suggesting that these two factors might compete for lysosomal gene regulation in these cells. TFEB ChIP-seq data obtained in HUVECs (GSM2354032) were then used to define a set of genes (n = 126) having TFEB-binding sites within ±2 kb from their TSSs and belonging to the lysosome gene set GO: 0005764. Examination of these TFEB lysosomal targets revealed that 25.4% (32 of 126) of them were down-regulated by the expression of TET2 CD (Fig 5G). Consistent with a role of MYC in shaping the chromatin landscape of these TFEB targets, MYC[high] TSSs of TFEB lysosomal target genes showed a strong decrease in H3K4me3 levels in TET2 CD cells and a slight increase in TET2 mCD cells, whereas MYC[low] TSSs of TFEB lysosomal targets did not show variations in H3K4me3 levels (Fig 5H). As expected, lysosomal gene mRNA levels were positively correlated with *TFEB* and *TFE3* levels and negatively correlated with MYC and TET2 levels in breast cancer patients (normal-like tumors, n = 639, Fig 5I). In addition, TET2[high]/MYC[high] tumors had lower levels of *CLN3*, *CTSD*, and *NAGLU* mRNAs

compared with TET2[high]/MYC[low] tumors (Figs 5J and S4F), validating in vivo the hypothesis that TET2 repression of lysosomal genes is dependent on MYC.

We next challenged MCF-7 clones with SRT1720, a synthetic compound activating SIRT1 and known to activate autophagy and enhance lysosomal membrane permeabilization, a process leading to the death of breast cancer cells (71). SRT1720 has also been shown to enhance TET2 enzymatic activity in myelodysplastic syndrome hematopoietic stem/progenitor cells (72). Thus, the impact of SRT1720 on global 5hmC levels in MCF-7 cells was first analyzed by dot blot. Data indicated that SRT1720, in MCF-7 cells, did not increase 5mC oxidation (Fig S4G), ruling out a possible regulation of TET activity by SRT1720 in these cells. At a concentration of 2.5 $\mu$M, SRT1720 induced the appearance of cytoplasmic vacuoles as soon as 4-h posttreatment and these vacuoles were cleared after 24 h in the EV and TET2 mCD cells whereas they remained visible, together with hyper-vacuolated dead cells, in TET2 CD cells (Fig 5K). After 4 d of treatment with daily doses of SRT1720, marked differences in cell survival were detected between clones, with a drastic reduction in viable TET2 CD cells compared with TET2 mCD cells (Fig 5L). In the presence of SRT1720, the addition of chloroquine, a drug that increases lysosome pH and inhibits fusion of autophagosomes with lysosomes (73), exacerbated the phenotype of TET2 CD cells which accumulated very large vacuoles (Fig S4H). In addition, serum starvation, a condition triggering autophagy through mTORC1 inhibition (74), induced high cell death rates in TET2 CD cells, whereas TET2 mCD cells, likely through a dominant-negative function of the inactive catalytic domain, were protected from death (Fig 5M and N). Collectively, these data indicated a prominent role of a TET2/MYC cross-talk in controlling lysosomal activity in breast cancer cells and impeding survival upon autophagy induction.

## Discussion

In vivo DNA methylation dynamics relies in part on the respective levels of enzymes having opposite roles, namely, DNMTs and TETs. Recent investigations using cell systems with combinatorial knockout of these enzymes and live-cell imaging of DNA methylation reporters provide direct evidence for a cyclical behavior of DNA methylation, with 5mC oxidation by TETs being a major contributor to the turnover of methylation at a genome-wide scale (75, 76, 77, 78). CpGs that appear highly methylated at the steady state

the *ESR1* gene encoding ERα is down-regulated both in decitabine-treated MCF-7 cells and in TET2 CD cells. **(C)** RT-qPCR validation of the similar and opposite effects of decitabine and TET2 CD expression on *MCAM* and *DSCR8* RNA levels, respectively (mean ± SEM, n = 3). **(D)** Functional annotation of TET2 CD and decitabine up-regulated genes with GREAT. Only the top significantly enriched GO Biological processes are shown. **(E)** Outline of the antiviral response pathway. Double-stranded (ds) RNAs originating from viruses or from transcription of endogenous retroviral sequences (ERVs) can be sensed by pattern recognition receptors (PRRs), which activate the mitochondria-associated protein MAVS. Active MAVS is protected from degradation by OTUD4 and stimulates the nuclear translocation of IRF3, IRF7, and NF-KB transcription factors to induce expression of type I interferon genes (IFNs) and, in turn, interferon-stimulated genes. Among interferon-stimulated genes, OAS1,2,3 and L activate RNAse L, which ultimately degrades dsRNAs. Activity of RNAse L can be counteracted by ABCE1 (figure made with BioRender; https://biorender.com).
**(F)** Heatmap representation of the fold change of genes implicated in the type I interferon and antiviral response pathways. **(G)** RT-qPCR measurement of *DDX58*, *OAS2*, and *IFIT1* in MCF-7 clones treated or not with 100 nM decitabine for 96 h (mean ± SEM, n = 3). **(H)** Correlation heatmap between TET2 and genes from the antiviral response pathway in BRCA tumors (normal-like tumors, n = 639). **(I)** RT-qPCR analysis of HERV-Fc1 and LTR12C endogenous retroviruses. Cells were treated with 100 nM decitabine for 96 h (mean ± SEM, n = 3). **(J)** Dot blot analysis of dsRNA in total RNA from EV, TET2 CD, and TET2 mCD cells. **(K)** RT-qPCR analysis of *ABCE1*, *DDX58*, *OAS1*, and *IFIT1* in MCF-7 cells transfected either with a scrambled (scr) siRNA or with siRNA B and C targeting *ABCE1* (mean ± SEM, n = 3). **(L)** Cell viability (MTT assay) after transfection of increasing concentrations of scrambled siRNA (scr) or *ABCE1* siRNA C (mean ± SEM, n = 3).

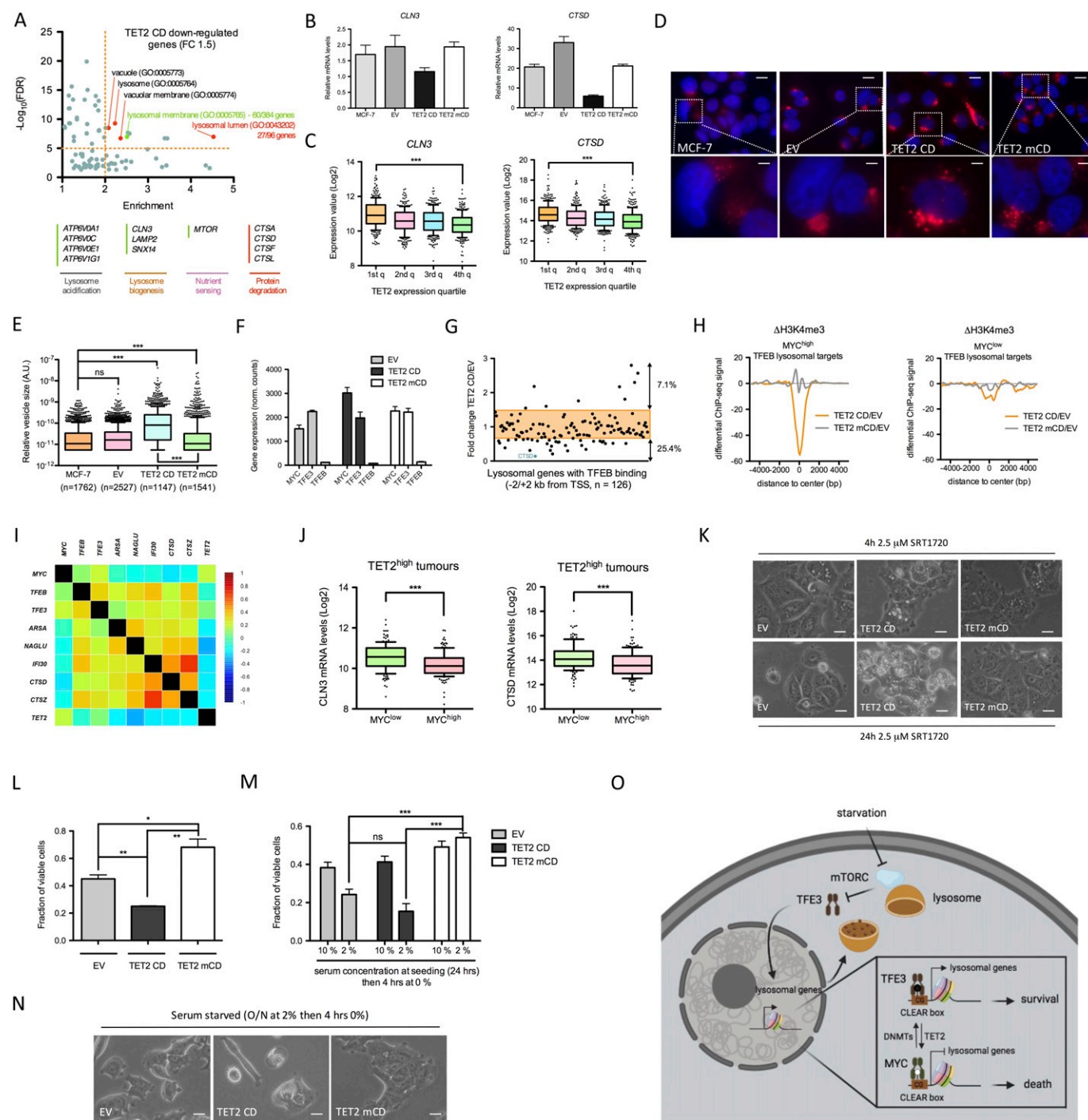

**Figure 5. TET2 alters lysosome function.**
**(A)** GO cellular component annotation (Pantherdb) of 1.5-fold down-regulated genes in TET2 CD cells compared with empty vector cells. Specific down-regulated genes from the lysosomal membrane and lysosomal lumen annotations are shown as examples. **(B)** RT-qPCR analysis of *CLN3* and *CTSD* mRNA levels in MCF-7 clones (mean ± SEM, n = 3). **(C)** Expression levels of *CLN3* and *CTSD* as a function of *TET2* mRNA levels ranked in quartiles (first quartile: lowest expression, fourth quartile: highest expression) in BRCA tumors (TCGA BRCA dataset, n = 1,218). **(D)** LysoTracker Red labeling of acidic vesicles in MCF-7 clones (scale bars: 20 $\mu$m for upper images and 5 $\mu$m for lower images). **(E)** Semiquantification of acidic vesicle size in MCF-7 clones with ImageJ. **(F)** Expression levels (RNA–seq–normalized read counts, mean ± SEM, n = 3) of transcription factors regulating lysosomal genes in MCF-7 clones. **(G)** Expression fold change in TET2 CD cells compared with empty vector cells of 126 genes associated with lysosome biogenesis and function and engaged by TFEB (ChIP-seq data from HUVECs) within 2 kb of their transcription start sites. **(H)** Differential H3K4me3 ChIP-seq signal at MYC[high] (left panel) and MYC[low] (right panel) transcription start sites of TFEB lysosomal target genes. **(I)** Correlation heatmap between transcription factors regulating lysosome biogenesis and lysosomal genes in BRCA tumors (normal-like tumors, n = 639). **(J)** *CLN3* and *CTSD* mRNA levels in TET[high] tumors (TCGA BRCA dataset, fourth quartile of expression, n = 302) ranked according to MYC expression (n = 151 for MYC[high] and MYC[low]). **(K)** Representative images of MCF-7 clones treated for 4 or 24 h with 2.5 $\mu$M SRT1720 (scale bars: 20 $\mu$m). **(L)** Cell viability (MTT assay) of the MCF-7 clones treated daily with 2.5 $\mu$M SRT1720 for 96 h (mean ± SEM, n = 3). **(M)** Cell viability

have low 5mC oxidation rate likely because they are poorly accessible to TETs. On the contrary, intermediate levels of methylation are reflecting a higher turnover, thanks to the engagement of TETs. This is particularly true with enhancers which are major spots of 5mC oxidation in the genome (47, 77, 79). Promoter CGIs are sites of nucleosome depletion and as such should be highly accessible to TETs. Accordingly, the CXXC domain-containing TET1 and TET3 accumulate at TSS-CGIs (16, 80), where they are believed to protect DNA from aberrant methylation by the DNMTs. In addition, supported by the observed gain in DNA methylation upon TET2 knockout at a substantial number of CGIs (81), TET2 most likely accumulates at TSS-CGIs. Although it does not contain a CXXC domain, TET2 could be targeted to TSS-CGIs through interaction with IDAX (82). Hence, promoter CGIs that are qualified as unmethylated in whole-genome bisulfite sequencing experiments could be protected against DNA methylation through a high 5mC oxidation rate. However, chromatin marks found at promoters (i.e., H3K4me3) can repress DNMT activity, providing an additional mechanism that could explain the lack of detectable DNA methylation at CGIs (83, 84). Here, by overexpressing either a catalytically active or inactive domain of TET2, we could highlight various operating modes of this protein. First, 5mC oxidation–dependent gene activation is observed at promoters that acquire H3K4 methylation with TET2 CD expression and not with TET2 mCD. Second, PRC2-associated gene repression (i.e., gain in H3K27me3) is partially 5mC oxidation–independent. Third, a number of gene repression events were mediated by an active 5mC oxidation mechanism as revealed by a decrease in H3K4me3 at TSSs in TET2 CD cells and an opposite regulation in TET2 mCD cells. These antagonistic effects of TET2 CD versus mCD could reflect the recruitment of a transcriptional repressor that binds to unmethylated sequences in TET2 CD cells, whereas demethylation would be impaired in TET2 mCD cells. We hypothesize that MYC could be one such factor because it is highly sensitive to DNA methylation (44), and engagement of MYC at TSSs induced either mild repression or activation (45). In favor of such cross-talk between TET2 and MYC, we show that TET2 CD expression is associated with demethylation of MYC-binding motifs and coordinately represses genes involved in lysosome biogenesis and function, a characteristic that has also been assigned to MYC (70). Interestingly, MYC competes at the TSS of lysosomal genes with the activation factors TFEB and TFE3 (70). Because TFEB is not expressed in MCF-7 cells, the lysosomal transcriptional program is likely to be activated by TFE3 in these cells. Notably, TFE3 has been shown to bind both unmethylated and methylated CACGTG sites in vitro (44), although a negative impact of DNA methylation on TFE3 binding to DNA was also described (39). Based on these observations, we propose a model that positions the antagonistic effects of DNMTs and TETs at the CLEAR motif as a central regulatory switch for fine-tuning of the lysosomal program (Fig 5O). This switch would operate not only in breast cancer tumors or in other human tumor types but also in other species for which the lysosomal program is controlled

through a competition between MYC and other bHLH-zip factors. In this regard, a remarkable enrichment of TET3 at TSSs of lysosomal genes (28% of the identified TET3 ChIP-seq peaks) in association with CACGTG motifs was described in the mouse brain (80), reinforcing the hypothesis of a widespread involvement of TETs in controlling lysosomal functions.

The coordinated down-regulation of lysosomal genes by TET2 CD, although of a low magnitude for each individual gene, is likely to trigger a lysosomal storage disease–like state in breast cancer cells. *CTSD* appeared as the most affected gene in this process and is a central actor of lysosomal activity. *CTSD* knockout mice develop a lysosomal storage disease that ultimately leads to death (85). In a mouse model of breast cancer, CTSD deficiency in the mammary epithelium impairs mTORC1 signaling and triggers the appearance of vacuolated cells with reduced proliferative activity upon serum starvation (86). TET2 CD cells were particularly prone to accumulate vacuoles, in particular when treated with the autophagy inducer SRT1720, and were highly sensitive to serum starvation. Cells respond to starvation by inhibiting lysosome-associated mTORC1, thus enhancing nuclear translocation of TFEB and TFE3 transcription factors and the activation of autophagy and lysosomal genes (87). Although an in-depth characterization of the impact of TET2 CD expression on these complex pathways will be required to fully understand the phenotype of these cells, we propose that TET2 CD weakens the lysosomal function and alters the cellular response to serum deprivation and autophagy induction. Our finding that TET2 expression, in the context of high MYC expression, negatively correlates with the mRNA levels of several lysosomal genes in breast tumors suggests that treatments combining autophagy inducers with DNA hypomethylating agents such as decitabine and/or TET-activating molecules could be beneficial to cancer patients. In this context, vitamin C, a compound that has TET-activating potential (88), could easily be administered to patients. Increasing TET protein levels in tumors, through anti-miR strategies against miRNAs targeting TET mRNAs (89), could provide an interesting alternative.

# Materials and Methods

### Cell culture and reagents

MCF-7 cells stably transfected with an EV or with plasmids encoding either the wild-type mouse TET2 catalytic domain (CD; aa916-1921, pcDNA3-Flag-TET2 CD, addgene #72219, (18)) or the catalytically dead mutant H1304Y, D1306A (pcDNA3-Flag-TET2 mCD, #72220; Addgene (18)) were grown in high-glucose and pyruvate-containing DMEM (41966; Gibco) supplemented with 10% fetal calf serum (S116365181H; Eurobio), nonessential amino acids (11140035; Gibco), penicillin–streptomycin (15240; Gibco), and Geneticin (11811064; Gibco) at 37°C and 5% $CO_2$. The SIRT1 activator compound SRT1720

---

(MTT assay) of the MCF-7 clones grown for 24 h in 10% or 2% serum and switched to serum-free medium for 4 h (mean ± SEM, n = 3). **(N)** Representative images of MCF-7 clones grown for 24 h in 2% serum and switched to serum-free medium for 4 h (scale bars: 20 μm). **(O)** Hypothetical model of the impact of TET2 on the coordinated transcription of lysosomal genes. Under starvation, TFE3 translocates to the nucleus where it activates lysosomal genes through binding to the CACGTG-containing CLEAR motif. Turnover of 5mC (black lollipop: 5mC, white lollipop: unmethylated C) at the CLEAR motif is controlled by the respective actions of DNMTs and TET2 and impacts the competitive binding of TFE3 and MYC, leading to an altered survival capability (figure made with BioRender).

was purchased from Sigma-Aldrich (567860), and chloroquine was from the CYTO-ID Autophagy Detection Kit 2.0 (ENZO ENZ-KIT175). Decitabine (5-aza-deoxycytidine) was from Sigma-Aldrich (A3656). For RT-qPCR analysis, cells were treated with 100 nM decitabine given every 36 h, for 96 h.

## Cell cycle analysis, migration, and clonogenic assays

The cell cycle was analyzed by flow cytometry. Briefly, 2,000,000 cells were plated on 10-cm dishes in DMEM supplemented with 10% FBS. After 72 h, the cells were trypsinized and fixed with 70% ethanol before being stained with propidium iodide in the presence of RNAse A. The cells were acquired on a Fortessa Becton Dickinson cytometer (Flow Cytometry, Biosit facility), and cell cycle analysis was performed with BD FACSDiva software. For 5-ethynyl-2'-deoxyuridine (EdU) labeling of S-phase cells, 250,000 cells were plated on coverslips in six-well plates. After 24 h, the cells were incubated with the serum- and steroid-deficient medium and then treated with 10 nM $E_2$ or vehicle for 24 h in a 0.5% serum-supplemented medium. Incorporated EdU was then fluorescently labeled with Alexa Fluor 488 by click chemistry (Click-iT EdU Imaging Kit; Invitrogen) according to the manufacturer's instructions. For wound healing assays, $650 \times 10^3$ cells were seeded into 10-mm$^2$ dishes. After overnight culturing, the cells were starved for 72 h. The confluent starved cells were wounded with a pipette tip and treated with E2 (10 nM) or ethanol. Images of recovery were captured daily after the renewal of the medium containing E2 (10 nM) or ethanol. For soft agar assays, 5,000, 10,000, or 20,000 cells were, respectively, seeded on soft agar 10-cm dishes (0.5% and 0.33% agar for bottom and top layers in complete medium, respectively). After 4 wk of culture (adding 500 $\mu$l of complete medium twice a week to avoid desiccation), colonies were stained with 0.005% crystal violet in 2% ethanol, imaged, and counted.

## siRNA transfection and cell viability determination

Levels of ABCE1 mRNAs were reduced by reverse transfection with 27-mer duplex siRNAs targeting ABCE1. Increasing concentrations (0, 2.5, 5, and 10 nM) of scrambled (scr, SR30004; OriGene) or ABCE1 siRNAs diluted in Opti-MEM (31985070; Thermo Fisher Scientific) were transfected in triplicates in 48-well plates with Lipofectamine RNAiMAX (13778075; Thermo Fisher Scientific) before seeding EV, TET2 CD, and TET2 mCD cells ($2.5 \times 10^4$ in each well). After 48 h, cell viability was quantified using an MTT detection kit (ab211091; Abcam). The medium was replaced by 100 $\mu$l of a 1:1 mix of phenol red–free DMEM without serum and a 10% MTT solution, and the plate was further incubated for 3 h at 37°C. The formed insoluble formazan crystals were then dissolved with a 1:1 solution of DMSO (D8418; Sigma-Aldrich) and isopropanol (33539; Sigma-Aldrich). After 30 min of incubation and a transfer into a 96-well plate, absorbance was detected at OD = 570 nm using a BioTek microplate reader (Power wave XS). For TET knockdown, 10 nM of siRNAs were reverse-transfected in MCF-7, T47D, and HEK293T cells as described earlier. After 48 h, total RNAs were extracted and subjected to RT-qPCR analysis. All siRNA sequences are listed in Table S2.

## Detection of acidic vesicles

Acidic vesicles were labeled with LysoTracker Red (L7528; Thermo Fisher Scientific). Cells were seeded on glass coverslips and grown in complete medium (DMEM, 10% serum) for 24 h before adding LysoTracker Red (Thermo Fisher Scientific) directly in the medium (final concentration: 75 nM) and Hoechst 33342 for nuclear staining. After 30 min at 37°C, the cells were washed once with PBS and fixed for 15 min with 4% paraformaldehyde in PBS. After two washes with PBS, coverslips were mounted on a glass slide with Vectashield (H-1200; Vector Laboratories). The dells were imaged with an Olympus BX-51 fluorescence microscope (60× magnification), and images were processed with ImageJ for quantification. Briefly, color channels were split, and the red channel images were processed with "Find edges" before threshold adjustment and particle size determination. Statistical differences (*t* test) were determined with Prism 5 (GraphPad Software Inc.).

## Dot blot detection of 5hmC

Genomic DNA (gDNA) was prepared using the DNeasy Blood & Tissue Kit (69506; QIAGEN). Relative 5hmC levels were quantified by blotting 500 ng of gDNA on a nitrocellulose membrane with an anti-5hmC antibody (39769; Active Motif) diluted 1/10,000, followed by an anti-rabbit antibody coupled to horseradish peroxidase (NA934; Dutscher) diluted 1/5,000. DNA was stained with 0.04% methylene blue in 0.5 M sodium acetate.

## RNA preparation and dot blot detection of dsRNAs

Total RNAs were extracted from 5.10$^7$ EV, TET2 CD, and TET2 mCD cells using TRIzol reagent (15596018; Thermo Fisher Scientific) according to the manufacturer's instructions. 2,000, 1,000, and 500 ng of each RNA sample were spotted on a nylon membrane (Hybond N, Dutscher RPN203B) previously soaked in 2× SSPE solution (0.3 M NaCl, 20 mM sodium phosphate, 2 mM EDTA) and inserted in the dot blot apparatus (SCIE-PLAS). RNAs were cross-linked to the membrane by 30-min heating at 80°C. The membranes were incubated overnight with an anti-dsRNA antibody (dsRNA mAb J2; Scicons) diluted 1/500, followed by an anti-mouse antibody coupled to horseradish peroxidase (sc-2005; Santa Cruz) diluted 1/10,000. Total RNAs were stained with 0.04% methylene blue in 0.5 M sodium acetate.

## RT-qPCR analyses

Total RNAs were isolated from $5 \times 10^7$ cells using TRIzol reagent (15596018; Thermo Fisher Scientific) according to the manufacturer's protocol. Reverse transcription was performed using 500 ng of total RNAs as template, 200 units of M-MLV reverse transcriptase (28025013; Thermo Fisher Scientific), and 250 ng of Pd(N)6 random hexamers (PM-301L; Euromedex). Real-time qPCR of reversed-transcribed RNAs was run with SYBR Green Master Mix (1725006CUST; Bio-Rad) in a Bio-Rad CFX96. Data were normalized to the positive control CDK8 using the $2^{-\Delta\Delta Ct}$ method. Primers were designed using Primer3 software (http://frodo.wi.mit.edu/primer3/; (90)) and were synthetized by Sigma-Aldrich. Primer sequences are listed in Table S2.

## ChIP-seq

All experiments were performed under hormonal depletion: Cells were kept for 48 h in phenol red–free DMEM (31052; Gibco) supplemented with 2.5% dextran/charcoal-treated fetal calf serum (S116365181W; Eurobio), glutamine, sodium pyruvate, nonessential amino acids, penicillin–streptomycin, and Geneticin at 37°C and 5% $CO_2$. For H3K4me3 and H3K27me3 ChIP-seq experiments, 4 million cells were fixed in 1.5% formaldehyde (F8775; Sigma-Aldrich) for 10 min at room temperature, and the reaction was stopped by the addition of glycine (100 mM). The cells were lysed in lysis buffer (150 mM Tris–HCl pH 8.1, 10 mM EDTA, 0.5% Empigen BB, 1% SDS, protease inhibitor cocktail) (5056489; Roche) and sonicated using a Bioruptor (15 min 30 s on/30 s off; Diagenode). The sonicated chromatin was incubated at 4°C overnight with either an anti-H3K4me3 (04-745; 1 µg; Millipore) or anti-H3K27me3 (07-449; 1 µg; Millipore) antibody in IP buffer (2.8 ng/ml yeast tRNA, 20 mM Tris–HCl, 2 mM EDTA, 150 mM NaCl, 1% Triton X-100, protease inhibitor cocktail). Complexes were recovered after incubation with 50 µl protein A–conjugated Sepharose bead slurry at 4°C. The beads were washed with washing buffers WB1 (20 mM Tris–HCl pH 8, 2 mM EDTA, 150 mM NaCl, 0.1% SDS, 1% Triton X-100), WB2 (20 mM Tris–HCl pH 8, 2 mM EDTA, 500 mM NaCl, 0.1% SDS, 1% Triton X-100), WB3 (10 mM Tris–HCl pH 8, 1 mM EDTA, 250 mM LiCl, 1% deoxycholate, 1% NP-40), and WB4 (10 mM Tris–HCl pH 8, 1 mM EDTA), and fragments were eluted with extraction buffer (1% SDS, 0.1 M NaHCO_3). For the preparation of each sequencing library (TruSeq; Illumina), ChIPed DNA from nine independent ChIP experiments were pooled and sequenced at the GenomEast Platform (IGBMC). Primers used for H3K27me3 ChIP-qPCR were synthetized by Sigma-Aldrich. The primer sequences are listed in Table S2.

## RNA-seq

RNA-seq was performed in triplicates on single EV, TET2 CD, and TET2 mCD clones after 4 h of treatment with $E_2$ (10 nM)/ethanol of cells previously maintained in phenol red–free medium supplemented with 2.5% charcoal-treated fetal calf serum during 48 h. Total RNA extraction was run using an RNeasy Plus kit (QIAGEN), which includes an optional DNAse digestion. For each condition, three replicate libraries (TruSeq stranded mRNA) were prepared and sequenced (single reads of 75 bases) at the Genomic Paris Centre facility. For RNA-seq statistical analysis, one replicate of the TET2-mCD control RNA-seq was ignored because of its lack of similarity with the other two samples of the triplicate, according to principal component analysis and hierarchical clustering.

## SCL-exo-seq

Selective chemical labeling-exonuclease (SCL-exo, (35)) experiments were conducted on starved cells after 50 min of E2 (10 nM)/ethanol treatment. Genomic DNA was extracted using the DNeasy Blood & Tissue Kit (69506; QIAGEN). For each experiment, 8 µg of gDNA was sonicated two times for 7 min (30 s on/30 s off) and two times for 14 min (30 s on/30 s off) with a Bioruptor (Diagenode). Glucosylation and biotinylation of 5hmC were performed with the Hydroxymethyl Collector kit (55013; Active Motif), followed by on beads-exonuclease digestion of the captured fragments and library preparation (TrueSeq library preparation kit, IP-202-1012; Illumina). For normalization purpose, 400 pg of 5hmC control DNA provided by the Hydroxymethyl Collector Kit was added to each sample as spike-in. Libraries from seven independent SCL-exo experiments were sequenced on seven lanes of a HiSeq 1500 (Illumina) by the GEH facility.

## Quantification and statistical analyses

RNA-seq reads were mapped to hg19 with Bowtie (91), and transcripts were quantified with RSEM (92). Differentially expressed genes were identified from RNA-seq data by the R package DESeq2 (93) after filtering the raw data using the R package HTSFilter (94). Online tools (GREAT, (95); http://bejerano.stanford.edu/great/public/html/; Panther, (38); http://www.pantherdb.org/) were used for interpretation and functional annotation. ChIP-seq reads were mapped to hg19 using Bowtie (91). SAMtools (96) generated bam files, which were processed with MACS (97) to generate wig files. Peak calling was followed a previously described procedure (79). Sequencing reads from publicly available datasets were mapped and treated following the same procedure as described earlier. MCF-7 MeDIP sequencing reads (DRA000030, (36)) and MYC ChIP-seq reads (SRR575112) were downloaded from https://ddbj.nig.ac.jp/resource/sra-submission/DRA000030 and from https://www.ncbi.nlm.nih.gov/sra?term=SRX188954, respectively. Differential wig files were generated by subtracting the signal in EV cells from the signal in either TET2 CD or TET2 mCD cells after normalization to the number of reads. SCL-exo reads were mapped separately on both strands with Bowtie (91). The resulting SAM files were processed with a Python script (https://mycore.core-cloud.net/index.php/s/4gyZ9dLTqgo86dt) to identify 5hmCpGs (98). Heatmaps were generated online with Cistrome ((41); http://cistrome.org/ap/root). ChIP-seq and SCL-exo data were normalized to input as follows: for every position of the wig file, a window of 100 bp surrounding that position was considered, and the input signal values in that window were averaged. Sample signal values were next divided by the mean input value of their corresponding window. For transcription factor motif search, bed files containing the coordinates of CpGs identified by SCL-exo were analyzed online with TFmotifView (42). The Cancer Genome Atlas (TCGA) RNA-seq data from breast cancer patients were downloaded from UCSC Xena (https://xenabrowser.net/). Heatmaps shown in Fig S4 were generated online by UCSC Xena (https://xenabrowser.net/heatmap/). RNA-seq data from intrinsic molecular breast cancer subtypes (PAM50) were interrogated with bc-GenExMiner v4.6 ((99); http://bcgenex.ico.unicancer.fr/BC-GEM/GEM-Accueil.php?js=1). Violin plots from Figs 1, 2, and S1 were generated online by bc-GenExMiner v4.6. Statistical differences were analyzed using Dunnett–Tukey–Kramer's test. Venn diagrams comparing gene lists were generated online (http://bioinformatics.psb.ugent.be/webtools/Venn/). Bar graphs were generated with GraphPad Prism 5.0 and analyzed by using the unpaired $t$ test (Prism 5.0). In each case, the number of samples is indicated in the corresponding figure legend.

# Data Availability

All sequencing data are available at GEO (accession code GSE173344, https://www.ncbi.nlm.nih.gov/geo/). Unnormalized wig files can be visualized at UCSC genome browser: https://genome.ucsc.edu/s/savner/MCF7_TET2. Flow cytometry data can be accessed at FlowRepository (https://flowrepository.org/) with the reference FR-FCM-Z446, reviewer access: https://flowrepository.org/id/RvFrDg4TjBSBZutxyIkLijvfWiuMcPcDaAiOmgl0wmqhoVhddiaSEbEbFDGxkFoF.

# Supplementary Information

# Acknowledgements

We are grateful to Romain Gibeaux for his help with microscopy experiments. We thank Rémy Le Guével (ImPACcell, Biosit, Rennes) for his help in setting up cell migration assays, and members of the GEH (Rennes) and GenomEast Platform (IGBMC, Strasbourg) sequencing facilities. This work was funded by the AVIESAN/Plan Cancer and La Ligue Contre le Cancer. The Genomic Paris Centre facility was supported by the France Génomique national infrastructure, funded as part of the "Investissements d'Avenir" program managed by the Agence Nationale de la Recherche (contract ANR-10-INBS-0009).

## Author Contributions

A Laurent: investigation and writing—review and editing.
T Madigou: investigation and writing—review and editing.
M Bizot: investigation and writing—review and editing.
M Turpin: investigation and writing—review and editing.
G Palierne: investigation and writing—review and editing.
E Mahe: investigation.
S Guimard: investigation.
R Metivier: conceptualization and writing—review and editing.
S Avner: data curation, software, and writing—review and editing.
C Le Peron: conceptualization, supervision, investigation, and writing—review and editing.
G Salbert: conceptualization, supervision, funding acquisition, investigation, and writing—original draft, review, and editing.

## Conflict of Interest Statement

The authors declare that they have no conflict of interest.

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
