## [Reviewer comments · Life Science Alliance]

Life Science Alliance

TET2-mediated epigenetic reprogramming of breast cancer cells impairs lysosome biogenesis

Audrey Laurent, Thierry Madigou, Maud Bizot, Marion Turpin, Gaëlle Paliarne, Elise Mahé, Sarah Guimard, Raphaël Métivier, Stéphane Avner, Christine Le Péron, and Gilles Salbert

DOI: <https://doi.org/10.26508/lsa.202101283>

Corresponding author(s): Gilles Salbert, University of Rennes 1 and Christine Le Péron, Université Rennes 1

Review Timeline:

Submission Date:	2021-10-29
Editorial Decision:	2021-11-02
Revision Received:	2022-02-02
Editorial Decision:	2022-03-04
Revision Received:	2022-03-11
Accepted:	2022-03-14

Transaction Report:

Please note that the manuscript was reviewed at *Review Commons* and these reports were taken into account in the decision-making process at *Life Science Alliance*.

Review
COMMONS

Review #1

The authors have done a very good job of analysing gene expression in MCF-7 cells overexpressing the TET2 catalytic domain (TET2CD) and its catalytically inactive mutant (mTET2CD). I have no technical issues with the manuscript: the experiments are carefully and rigorously performed and the results are interpreted thoughtfully. The manuscript analyses a system that is not entirely physiological, but the experiments are very carefully performed and interpreted and so I personally feel that the study makes a significant contribution to the literature. I have just one question and two minor comments:

1. In Figure 5, the relation of TET2CD overexpression to DNA methylation status and gene expression is not clear. Overexpression of TET2CD in MCF-7 cells resulted in increased expression of genes involved in antiviral responses and interferon-induced genes (encoding RIG-I, MDA5, LGP2, OAS1/2/3/L, IFIT1/2/3/5, IFI27). One would assume this would be due to TET-mediated demethylation of their regulatory regions, but only some of these genes showed increased expression in cells treated with decitabine, which would also presumably lead to demethylation of regulatory regions. There is some evidence for a common mechanism of action, however, since there was a strong synergistic increase in gene expression for at least three of these genes (encoding RIG-I, OAS2 and IFIT1) when TET2CD-expression and decitabine treatment were combined. This was also true for the endogenous retrovirus HERV-Fc1. In each case, there was either a partial or complete requirement for TET catalytic activity, since the increase in expression was lower in cells expressing mTET2CD. Can the authors explain their thoughts and conclusions more clearly, either in the text or in the discussion?
2. Minor point: On p. 9, line 6: "(ii) those having readily detectable 5hmC in EV cells and losing 5hmC in TET2 CD cells (n=95), likely due to further oxidation of 5hmC and replacement by unmodified C". This statement is only partially accurate, since even if the further oxidised forms 5fC and 5caC were not excised, they would still be read as unmodified C after bisulphite treatment. Please modify the statement to be more accurate.
3. Minor point: The citations in the text are inconsistent, with some citations listed as (Name) et al., some as numbered references, and occasionally as both. Please correct.

Review #2

The manuscript by Laurent et al. describes how enforced expression of the catalytic domain of the TET2 methylcytosine dioxygenase enzyme in the MCF7 breast cancer cell line decreases its tumorigenic potential (reduced anchorage-independent growth and cell migration). In addition, it is shown that TET2 overexpression results in sensitivity to autophagy inhibitors by increasing MYC occupancy at regulatory regions of lysosomal genes, resulting in their repression.

The article presents a detailed exploration of several potential mechanisms for an anti-tumorigenic activity of TET2 CD overexpression. Impact on gene transcription occurs in part via modulation of methylation but also in part in a catalytic activity independent manner. Some of the target genes identified are compatible with the observed reduced cell growth and migration. Reduced estrogen receptor expression, binding to DNA and regulation of target genes is also reported. Both activation and repression of CpG island genes was observed, but was not predicted by oxidation rates but rather by recruitment of H3K4me3 or H3K27me3 marks. Finally, the authors focus on a panel of MYC repressed genes involved in lysosome biogenesis, and show that TET2 overexpression results in increased sensitivity to autophagy inhibitors.

The results are well presented and their description is clear. Statistically relevance is presented and the authors use additional public data to provide relevance of results obtained in MCF7 cells to breast tumors. The article presents original findings and characterizes the phenotype of overexpressing TET2 CD in a comprehensive manner using a whole panel of genomic assays (RNAseq, ER and histone modification ChIP-seq, SLC-exo). Interesting observations include the fact that TET2 CD overexpression results in effects both on gene activation and repression, differing from decitabine treatment. Directionality of regulation is not predicted by oxidation dynamics, possibly due to a role of the TET2 CD in gene repression by recruitment of PRC2. The observed impact on lysosomal gene expression via effects on demethylation at MYC sites, is the most original conclusion of the paper and may have clinical implications due to increased sensitivity to autophagy inducers.

The main shortcoming of this paper is the fact that all observations and analyses were made in one cell line.

Verifying the main finding that demethylation of MYC sites leads to down-regulation of lysosomal biogenesis in different cell lines would boost confidence that these results have a general importance. The article also goes in many different directions, giving the impression of lack of focus. The observed reduction in estrogen receptor signaling expression is of potential interest, but no mechanistic conclusion is reached. This section should be condensed or removed.

Potential therapeutic applications are discussed but seem far removed. TET2 CD OE may differ from treatment with drugs activating TET. A discussion of whether other TETs have a similar mode of action would also be of interest in this respect.

November 2, 2021

Re: Life Science Alliance manuscript #LSA-2021-01283

Dr Gilles Salbert
Université de Rennes

Dear Dr. Salbert,

Thank you for submitting your manuscript entitled "TET2-mediated epigenetic reprogramming of breast cancer cells impairs lysosome biogenesis" to Life Science Alliance. We invite you to re-submit the manuscript, revised according to your Revision Plan.

Thank you for this interesting contribution to Life Science Alliance. We are looking forward to receiving your revised manuscript.

Sincerely,

- A letter addressing the reviewers' comments point by point.
- An editable version of the final text (.DOC or .DOCX) is needed for copyediting (no PDFs).
- High-resolution figure, supplementary figure and video files uploaded as individual files: See our detailed guidelines for preparing your production-ready images, <https://www.life-science-alliance.org/authors>
- Summary blurb (enter in submission system): A short text summarizing in a single sentence the study (max. 200 characters including spaces). This text is used in conjunction with the titles of papers, hence should be informative and complementary to the title and running title. It should describe the context and significance of the findings for a general readership; it should be written in the present tense and refer to the work in the third person. Author names should not be mentioned.
- By submitting a revision, you attest that you are aware of our payment policies found here: <https://www.life-science-alliance.org/copyright-license-fee>

B. MANUSCRIPT ORGANIZATION AND FORMATTING:

*****IMPORTANT:** It is Life Science Alliance policy that if requested, original data images must be made available. Failure to provide original images upon request will result in unavoidable delays in publication. Please ensure that you have access to all original microscopy and blot data images before submitting your revision. *******

1. General Statements

We would like to thank the reviewers for their positive comments and suggestions on how to improve our manuscript. This work was aimed at uncovering molecular mechanisms associated with variations in TET2 expression levels in breast cancer. Comparison of TET2-mediated effects and the effects of a chemically induced DNA demethylation (decitabine treatment) also provided useful information in defining TET2-specific events. Importantly, although data were generated through artificially modulating TET2 levels, the similarity between results observed in our artificial system and in primary tumors indicates that the mechanisms described here are likely to operate in cancer.

2. Point-by-point response to Reviewer's comments

REVIEWER #1

“Reviewer #1 (Evidence, reproducibility and clarity):

The authors have done a very good job of analysing gene expression in MCF-7 cells overexpressing the TET2 catalytic domain (TET2CD) and its catalytically inactive mutant (mTET2CD). I have no technical issues with the manuscript: the experiments are carefully and rigorously performed and the results are interpreted thoughtfully.

Reviewer #1 (Significance):

The manuscript analyses a system that is not entirely physiological, but the experiments are very carefully performed and interpreted and so I personally feel that the study makes a significant contribution to the literature.”

We thank the Reviewer for this positive assessment of our work.

“I have just one question and two minor comments:

1. In Figure 5, the relation of TET2CD overexpression to DNA methylation status and gene expression is not clear. Overexpression of TET2CD in MCF-7 cells resulted in increased expression of genes involved in antiviral responses and interferon-induced genes (encoding RIG-I, MDA5, LGP2, OAS1/2/3/L, IFIT1/2/3/5, IFI27). One would assume this would be due to TET-mediated demethylation of their regulatory regions, but only some of these genes showed increased expression in cells treated with decitabine, which would also presumably lead to demethylation of regulatory regions. There is some evidence for a common mechanism of action, however, since there was a strong synergistic increase in

gene expression for at least three of these genes (encoding RIG-I, OAS2 and IFIT1) when TET2CD-expression and decitabine treatment were combined. This was also true for the endogenous retrovirus HERV-Fc1. In each case, there was either a partial or complete requirement for TET catalytic activity, since the increase in expression was lower in cells expressing mTET2CD. Can the authors explain their thoughts and conclusions more clearly, either in the text or in the discussion? "

The fact that TET2 can act on both DNA and RNA could provide a possible explanation for the observed additive (rather than synergistic, to our point of view) effect of TET2 CD expression and decitabine treatment. Indeed, when considering that RNA harboring m5C is less prone to induce an antiviral response, and knowing that ERV RNAs are methylated in cells, we believe that m5C oxidation by TET2 CD (and not by TET2 mCD) in these RNAs could foster the antiviral response. This has been added in the Results section, page 6: "Knowing that (i) viral and ERV RNAs are methylated in cells (57,58), (ii) RNA methylation decreases the anti-viral response (59), and (iii) in ES cells TET2 oxidizes 5mC in ERV RNAs (58), it is then possible that the high anti-viral response in TET2 CD cells reflects a dual action of TET2 on both genomic DNA and RNA transcribed from repeated sequences. Such a scenario would be compatible with the additive effect observed when combining active TET2 CD and decitabine (Fig 4G,I)."

"2. Minor point: On p. 9, line 6: "(ii) those having readily detectable 5hmC in EV cells and losing 5hmC in TET2 CD cells (n=95), likely due to further oxidation of 5hmC and replacement by unmodified C". This statement is only partially accurate, since even if the further oxidised forms 5fC and 5caC were not excised, they would still be read as unmodified C after bisulphite treatment. Please modify the statement to be more accurate."

Since we used selective chemical labeling coupled to exonuclease digestion to map 5hmC, no bisulfite was used. However, we agree with the Reviewer that the absence of 5hmC in these particular regions cannot be formally attributed to replacement of oxi-mCs by unmodified cytosines. We have now changed this sentence to "likely due to further oxidation of 5hmC by the expressed TET2 CD".

"3. Minor point: The citations in the text are inconsistent, with some citations listed as (Name) et al., some as numbered references, and occasionally as both. Please correct."

We thank the Reviewer for alerting us on this problem. This has been corrected.

REVIEWER #2

"Reviewer #2 (Evidence, reproducibility and clarity):

The manuscript by Laurent et al. describes how enforced expression of the catalytic domain of the TET2 methylcytosine dioxygenase enzyme in the MCF7 breast cancer cell line decreases its tumorigenic potential (reduced anchorage-independent growth and cell migration). In addition, it is shown that TET2 overexpression results in sensitivity to autophagy inhibitors by increasing MYC occupancy at regulatory regions of lysosomal genes, resulting in their repression.

The article presents a detailed exploration of several potential mechanisms for an anti-tumorigenic activity of TET2 CD overexpression. Impact on gene transcription occurs in part via modulation of methylation but also in part in a catalytic activity independent manner. Some of the target genes identified are compatible with the observed reduced cell growth and migration. Reduced estrogen receptor expression, binding to DNA and regulation of target genes is also reported. Both activation and repression of CpG island genes was observed, but was not predicted by oxidation rates but rather by recruitment of H3K4me3 or H3K27me3 marks. Finally, the authors focus on a panel of MYC repressed genes involved in lysosome biogenesis, and show that TET2 overexpression results in increased sensitivity to autophagy inhibitors.

The results are well presented and their description is clear. Statistical relevance is presented and the authors use additional public data to provide relevance of results obtained in MCF7 cells to breast tumors.

Reviewer #2 (Significance):

The article presents original findings and characterizes the phenotype of overexpressing TET2 CD in a comprehensive manner using a whole panel of genomic assays (RNAseq, ER and histone modification ChIP-seq, SLC-exo). Interesting observations include the fact that TET2 CD overexpression results in effects both on gene activation and repression, differing from decitabine treatment. Directionality of regulation is not predicted by oxidation dynamics, possibly due to a role of the TET2 CD in gene repression by recruitment of PRC2. The observed impact on lysosomal gene expression via effects on demethylation at MYC sites, is the most original conclusion of the paper and may have clinical implications due to increased sensitivity to autophagy inducers.”

We thank the Reviewer for this positive assessment of our work.

“The main shortcoming of this paper is the fact that all observations and analyses were made in one cell line. Verifying the main finding that demethylation of MYC sites leads to down-regulation of lysosomal biogenesis in different cell lines would boost confidence that these results have a general importance.”

“A discussion of whether other TETs have a similar mode of action would also be of interest in this respect.”

The anti-correlation between TET2 expression and lysosomal gene expression is observed with pan-cancer patient data in addition to breast cancer data, suggesting that the proposed mechanism operates in various tissue types. Nonetheless, we agree with the reviewer that additional experiments in other cell lines could bring additional evidence that downregulation of lysosomal genes by TET2 (through demethylation of MYC binding sites) can be generalized. Since generating and characterizing clones ectopically expressing functional TET2 CD in various cell lines would have required months of intensive manipulation, we have rather run knock-down experiments in which individual TETs (i.e. TET1, TET2, and TET3) have been targeted by siRNAs in various cell lines, including MCF-7, HeLa, MDA-MB-231, T47D, LNCaP and HEK293T. Data obtained with MCF-7, T47D and HEK293T cells are now shown Fig. S4 and indicate that, as expected from our TET2 overexpression data showing a reduction in CTSD and CLN3 mRNA levels, CTSD and CLN3 expression levels were increased upon TET2 knock-down in MCF-

7, T47D and HEK293T cells, whereas they were not increased with TET1 or TET3 siRNA transfection. This suggests that TET2 specifically drives repression of lysosomal genes in these cells. Unfortunately, the other tested cells lines did not prove useful in this context since: (i) HeLa cells were found to express extremely low levels of TET1, TET2 and CTSD, preventing their use in this study, (ii) efficient knock-down of TET2 mRNA levels in MDA-MB-231 cells could not be obtained with the 2 siRNAs we used (18% and 22% reduction in TET2 mRNA levels), and (iii) LNCaP cells did not survive siRNA transfection conditions.

“The article also goes in many different directions, giving the impression of lack of focus. The observed reduction in estrogen receptor signaling expression is of potential interest, but no mechanistic conclusion is reached. This section should be condensed or removed.”

We agree with the Reviewer that no clear conclusion/explanation could be reached on the impact of TET2 expression on estrogen signaling and cell identity. Following the Reviewer’s recommendation, we have deleted this section and the associated Fig. 2 and Supplementary Fig. 2.

“Potential therapeutic applications are discussed but seem far removed. TET2 CD OE may differ from treatment with drugs activating TET.”

We agree that this point was only very briefly mentioned in the manuscript. Although we believe that discussing therapeutic strategies is somehow out of the scope of this manuscript, we added the following sentences at the end of the discussion section: “In this context, Vitamin C, a compound that has TET activating potential (87) could easily be administered to patients. Raising TET protein levels in tumours, through anti-miR strategies against miRNAs targeting TET mRNAs (88), could provide an interesting alternative.”

March 4, 2022

RE: Life Science Alliance Manuscript #LSA-2021-01283R

Prof. Gilles Salbert
University of Rennes 1
Institut de Génétique et Développement de Rennes
Campus de Beaulieu
263 avenue Général Leclerc
Rennes 35042
France

Dear Dr. Salbert,

Thank you for submitting your revised manuscript entitled "TET2-mediated epigenetic reprogramming of breast cancer cells impairs lysosome biogenesis". We would be happy to publish your paper in Life Science Alliance pending final revisions necessary to meet our formatting guidelines. Please address Reviewer 2's final comments.

- please add the Twitter handle of your host institute/organization as well as your own or/and one of the authors in our system
- please add ORCID ID for secondary corresponding author-they should have received instructions on how to do so
- we encourage you to revise the figure legend for Figures S1 such that the figure panels are introduced in alphabetical order
- please indicate molecular weight next to each protein blot
- please add a callout for Figure S2G to your main manuscript text
- please add scale bars to Figures 5D, 5K, 5M and S4H, and indicate their size in the Figure Legend

A. FINAL FILES:

B. MANUSCRIPT ORGANIZATION AND FORMATTING:

Sincerely,

Reviewer #1 (Comments to the Authors (Required)):

This is a revised version, so all I need to say is that the authors have addressed my concerns adequately and the revised manuscript is now suitable for publication in Life Science Alliance

Reviewer #2 (Comments to the Authors (Required)):

My main comments have been adequately addressed. I find the manuscript improved in focus by removal of the section on estrogen receptor, and in relevance by the inclusion of assays in additional cell lines. Some sections are still hard to read, particularly the one on the impact of TET2 CD on MYC target genes. This section could benefit from an introduction on the pleiotropic roles of Myc in transcriptional regulation, implicating repression as well as activation (e.g., ref 83), before arriving to the model that increased MYC binding upon TET2 CD over-expression leads to a lower transcription of the associated genes. However, globally this is a very interesting and original paper with relevance to the mechanistic interplay between epigenetic and cancer, and how it may be exploited for cancer treatment.

March 14, 2022

RE: Life Science Alliance Manuscript #LSA-2021-01283RR

Prof. Gilles Salbert
University of Rennes 1
Institut de Génétique et Développement de Rennes
Campus de Beaulieu
263 avenue Général Leclerc
Rennes 35042
France

Dear Dr. Salbert,

Thank you for submitting your Research Article entitled "TET2-mediated epigenetic reprogramming of breast cancer cells impairs lysosome biogenesis". It is a pleasure to let you know that your manuscript is now accepted for publication in Life Science Alliance. Congratulations on this interesting work.

DISTRIBUTION OF MATERIALS:

Again, congratulations on a very nice paper. I hope you found the review process to be constructive and are pleased with how the manuscript was handled editorially. We look forward to future exciting submissions from your lab.

Sincerely,
